# Unveiling the Emission and Variation Mechanism of Mrk 501: Using the Multi-Wavelength Data at Different Time Scale

Lizhi Liu [1,2], Yunguo Jiang [1,2,*], Junhao Deng [1,2], Zhaohao Chen [1,2] and Chenli Ma [1,2]

1 Shandong Provincial Key Laboratory of Optical Astronomy and Solar-Terrestrial Environment, Institute of Space Sciences, Shandong University, Weihai 264209, China; 202217736@mail.sdu.edu.cn (L.L.); 202117739@mail.sdu.edu.cn (Z.C.); 202017700@mail.sdu.edu.cn (C.M.)
2 School of Space Science and Physics, Shandong University, Weihai 264209, China
* Correspondence: jiangyg@sdu.edu.cn

**Abstract:** Variability study at multi-frequency provides us with rich information of the emission and variation mechanism for blazars. In this work, we present a comprehensive multi-frequency analysis of the high-synchrotron-peaked (HSP) blazar Mrk 501, using $\gamma$-ray, X-ray, optical, optical polarization, and radio data. The multiple-wavelength light curves are analyzed by using the localized cross-correlation function to derive locations of their emitting regions. The X-ray, $\gamma$-ray, and optical emitting regions are found to be upstream of the radio core region, while the X-ray and $\gamma$-ray emitting regions likely coincide. We studied the variation behaviors for three long-term (years), five relatively short-term (months) periods. We find a positive correlation between the optical and X-ray fluxes, and conclude that the variable of Doppler factor is not favored for the one-zone SSC scenario. The study also identifies the existence of a soft $\gamma$-ray background in the low-activity state, which could be explained by the spine/layer jet model. Our study on Mrk 501 provides valuable insights to understand the emission processes and variation mechanism for HSP blazars.

**Keywords:** galaxies; active—BL Lacertae objects; individual; Mrk 501—accleration of particles—$\gamma$-rays; galaxies—X-rays; galaxies

## 1. Introduction

BL Lacertae objects (BL Lacs) are a subclass of blazars, whose observed luminosity exceeds that of their host galaxies. Blazars make up the extreme subclass of radio-loud Active Galactic Nuclei (AGNs). The main radiation of BL Lacs is thought to originate from relativistic jets, which are viewed within the angles $\lesssim 10\,^{\circ}$ relative to the observer's line of sight (LOS) [1]. The variability of BL Lacs may originate from inherent changes in the relativistic jet. One can observe these fluctuations on all accessible timescales ranging from a few minutes to days, months, and even decades. BL lacs have emission at all wavelengths in the whole electromagnetic spectrum, ranging from radio to very high energy (VHE) $\gamma$-rays [2].

Jet-dominated AGNs exhibit a distinctive spectral energy distribution (SED) characterized by two distinct bumps. The lower-energy peak, spanning from radio to optical/X-ray frequencies, is generally attributed to synchrotron radiation emitted by relativistic electrons accelerated in the jet, as supported by the detection of significant linear polarization [3]. Based on the frequency at which the peak occurs ($\nu_{\text{peak}}$), blazars can be classified as the low-synchrotron-peaked (LSP; $\nu_{\text{peak}} \leq 10^{14}$ Hz), intermediate-synchrotron-peaked (ISP; $10^{14}$ Hz $\leq \nu_{\text{peak}} \leq 10^{15}$ Hz), or the high-synchrotron-peaked (HSP; $\nu_{\text{peak}} > 10^{15}$ Hz) sources [4]. In modeling the SEDs of jet-dominated AGNs, the most commonly used model is the one-zone leptonic model, which postulates that all of the jet's non-thermal emission originates from a compact region.

In the 1970s, Markarian and Lipovetskij [5] discovered Mrk 501 (z = 0.033, Pushkarev et al. [6]) and it was detected in very high energy (VHE) $\gamma$-ray with energies greater than

300 GeV by the Whipple Observatory Gamma Ray Collaboration after two decades [7]. The broadband emission from radio to $\gamma$-ray of Mrk 501 is dominated by non-thermal radiation that is produced in the innermost part of the jets, which are oriented very close to our line of sight [1]. The variation phenomena exhibited by Mrk 501, including their color index behavior, correlations between time series in various energy bands, and polarization fluctuations, have been extensively investigated throughout history. Xiong et al. [8] monitored the BL Lac object Mrk 501 in the optical *V*, *R*, and *I* bands from 2010 to 2015. The results show intra-day and long-term variability, with a dominant bluer when brighter (BWB) trend on intermediate, short-term, and intra-day time scales, supporting the shock-in-jet model. MAGIC collaboration et al. [9] studied the correlation between different bands using the analysis method of the discrete correlation function (DCF) and found that $\gamma$-ray has excellent correlation with X-ray. The research work on the polarization of Mrk 501 is also widely concerned by researchers of blazars, especially since the observation of X-ray polarization by *IXPE* observation has been carried out, and the study of polarization is of great help to constrain radiation mechanism. Liodakis et al. [10] report on the detection of X-ray polarization from Mrk 501. Their results suggest that the shock front is the source of particle acceleration, and also imply that the plasma becomes increasingly turbulent with distance from the shock. The study of multi-band observations and correlation analysis in different bands has become one of the primary research methods for investigating the mechanisms and physical processes of active galactic nucleus radiation.

During the multi-wavelength (MWL) campaign conducted in July 2014, a flaring activity lasting approximately two weeks was observed in both the X-ray and very-high-energy (VHE) bands. MAGIC Collaboration et al. [11] discusses the extreme X-ray outburst of the Mrk 501 galaxy during a certain period. The study found that there is correlation between X-ray and VHE radiation, while VHE radiation shows greater variability. This indicates that variability is caused by the acceleration and cooling of high-energy electrons. On the day with the highest X-ray flux, a narrow peak of about 3 TeV was observed in the VHE spectrum, which cannot be explained by traditional analytical functions i.e., a single-zone SSC model. Hu and Yan [12] also investigated the phenomenon of a narrow peak of about 3 TeV observed during that period. They proposed that synchrotron photons generated by a power-law distribution interact with electrons in the pileup distribution via inverse-Compton (IC) scattering to form the narrow spectral feature observed in the TeV energy range. For MJD 56700 to MJD 56950, the radiation mechanism in $\gamma$-ray during this period is more intricate and necessitates more comprehensive and detailed investigation to elucidate.

In this work, we aim to understand the location of emitting regions by studying the correlations between different light curves. We also investigate the phenomena of variations, including the color index, $\gamma$-ray spectral index at different timescales to infer the variaiton mechanism. The correlation analysis between the optical polarization degree and optical flux, as well as the $\gamma$-ray and X-ray fluxes are presented to reveal the primary variable. The SED analysis also help to verify the emission mechanism. This paper is organized as follows. In Section 2, we describe the process of collecting and reducing the data. In Section 3, we present a multi-frequency analysis of the blazar Mrk 501 using local cross-correlation function (LCCF) analysis to determine the locations of the emitting regions. In Section 4, we examined the correlations among various parameters, including the X-ray flux and $\gamma$-ray flux, the optical color index and magnitude, and the optical flux and polarization degree. For Section 5, we performed fitting of the broad-band spectral energy distribution (SED) using a one-zone model and proposed the use of the Spine/Layer Jet model to explain the soft trend observed in the low-energy part of the gamma-ray emission during the low-activity state. Our conclusion is given in Section 6.

## 2. Data Collection

The multiple-wavelength data of the target include the $\gamma$-ray data of Fermi-LAT, the X-ray data from Swift-XRT, the optical data of the Steward Observatory (SO), the radio 15-GHz data of Owen Valley Radio Observatory (OVRO), collected from public data archives.

**Fermi LAT data**: We collected nearly 14 years (from 4 August 2008 to 6 November 2022) $\gamma$-ray data of Mrk 501 from the public Fermi Science Support Center (FSSC)[1]. We set the search radius as $15°$ around the target. The Fermi-LAT data of the target in the energy interval of 0.1–300 GeV were analyzed by with the ScienceTools package version *v11r5p3*. In this pipeline, we consider the Fermi-LAT 12-year catalogue (4FGL-DR3, Abdollahi et al. [13]) together with the diffuse Galactic and isotropic backgrounds, namely gll iem v07.fits and *iso P8R3_SOURCE_V3V1.txt*, as well as the instrument response functions *P8R3_SOURCE_V3*. The unbinned likelihood algorithm is employed to derive the $\gamma$-ray spectrum and fluxes. We choose to use evclass = 128, evtype = 3 in the step of gtlike. We use the LogParabola spectral model for Mrk 501, with 4FGL name J1654.8+3945. The parameters for point sources located within a $5°$ radius from the ROI center were treated as free variables, while those for others were fixed using the values provided in the 4FGL-DR3 catalog. Counting the data points and the uncertainties, we choose the energy range of 0.9 Gev to 2.7 Gev to represent the $\gamma$-ray light curve. We opt for a time bin of 7 days, yielding 473 data points. In addition, we also used the *easyFermi*[2] tool as a data processing tool for the period from MJD 56,700 to MJD 56,950, because employing easyFermi for SED construction is more efficient in time. For the energy spectrum, we divided the energy range of 0.1–300 GeV into 10 logarithmically equal energy bins. Data points with TS < 10 were treated as 95% CL upper limits in both the light curve and the energy spectrum.

**X-ray data**: The X-ray data covering the period from 13 January 2006 to 3 June 2020, were gathered through the Swift-XRT[3] monitoring program, supported by the Fermi GI program and the Swift Team [14]. The energy range of the data spans from 0.3 to 10 keV. However, part of this data is taken in PC mode, which makes the counts/s not always trustable. For the energy spectrum, the target information was obtained from an area within a radius of 20 pixels (approximately 46 arcseconds), and the background level is estimated by selecting a ring-shaped region near the source with an inner and outer radius of 40 and 50 pixels for WT data. To generate the ancillary response files (ARFs), the xrtmkarf task is used. All data were analyzed using HEASOFT 6.28 starting from Level 1. The raw data was processed with the *XRT* pipeline to obtain the Level II data, including clean event files. Spectra were obtained using xselect on the Level II data from the photon counting mode and windowed timing mode. The spectra were grouped using grppha, with a minimum of 20 photons per bin for the spectra of the WT mode. Finally, in XSPEC program, we used the tbabs*po model for fitting, and divided the energy range of 0.3–10 keV into 10 energy bands.

**Photometry and polarization data**: The data of Photometry and polarization were retrieved from the long-term monitoring program operated by the Steward Observatory (SO)[4]. The Steward Observatory has monitored the target from 6 October 2008 to 7 July 2018, which includes the optical V-band, R-band, and polarization data. For the photometry and polarization data, we did not perform host galaxy correction.

**Radio 15 GHz data**: We collected nearly 11 years of data from the OVRO[5] 40 m monitoring program [15]. The data of 15GHZ were obtained from 22 January 2009 to 16 December 2020.

## 3. Location of Emitting Regions

### 3.1. Correlation Analysis

The analysis of time lags among multi-frequency time series proves to be an effective method for pinpointing the emitting region at a specific wavelength [16,17]. Correlations among unevenly sampled light curves can be examined using DCF [18]. However, it might produce spurious signals, potentially leading to a misinterpretation of the location of emitting regions [19]. Besides, the coefficients of the DCF can go beyond the range [−1, 1], making it not applicable to standard statistical tests [20]. On the contrary, the local

cross-correlation function (LCCF) has a good property for estimation of the significance level [21,22]. So, we use the LCCF to perform the correlation analysis between multiple wavelength light curves in this work.

We employ the Monte Carlo (MC) procedure to assess the significance of time lags. We simulate 10,000 artificial light curves by the method of Timmer and Koenig [23]. We calculate the spectral slope of the power spectral density (PSD) for the light curve data collected. Then, the spectral slope of PSD of the X-ray is calculated to be $fi_X = 1.0$, and that of the radio is $fi_{radio} = 0.96$. The simulated light curve of X-ray contains 5700 data points and the simulated radio light curve contains 4500 data points. Both of them have the time bin of one day. After simulating the light curve with bin 1 day, we will select the data points with exact the same sampling as the observed LC. This operation will eliminate the sampling effects in the significance estimation. Then, LCCFs between these artificial light curves and the observed light curves are calculated in order to obtain the distribution of LCCFs. We acquire confidence levels of $1\sigma(68.27\%)$, $2\sigma(95.45\%)$, and $3\sigma(99.73\%)$ to demonstrate the significance of correlation signals.

### 3.2. Time Lag Analysis

We calculate two kinds of time lags, $\tau_p$ and $\tau_c$. $\tau_p$ is the lag for the highest peak of LCCF, $\tau_c$ is the centroid lag around the peak of the LCCF, which is defined as $\tau_c = \Sigma_i \tau_i C_i / \Sigma_i C_i$, where $C_i$ is the correlation coefficient satisfying $C_i > 0.8 LCCF(\tau_p)$ [24]. This step is repeated 10,000 times to obtain the distribution of $\tau_p$ and $\tau_c$. We treat the standard deviation of $1\sigma$ as the error of time lags, which is calculated by using the flux randomization (FR) and the random subset selection (RSS) MC method [25,26].

In Figure 1, panels (a), (b), (c), and (d) display the LCCF results for the optical versus radio, X-ray versus radio, the optical versus X-ray, and $\gamma$-ray versus X-ray, respectively. Considering the sampling cadence, we set the time lag range for the LCCF analysis to be $[-1000, 1000]$ in panels (a), (b), and (c), and the lag bin is 10 days. Due to a 7-day time binning for the $\gamma$-ray light curve, considerations were made for days without data points. Additionally, it was ensured that the time delay binning was larger than the sampling rate of the light curve to avoid potential distortion of results. This precaution was taken to ensure the accuracy and validity of the obtained outcomes. While in panel (d), the time lag range is defined as $[-200, 200]$, with a lag bin size of 2 days. As shown in Figure 1, the peaks of LCCF in cases of the optical versus radio, X-ray versus radio, and optical versus X-ray are roughly beyond the $2\sigma$ significance level, and the peak of LCCF in the case of $\gamma$-ray versus X-ray is beyond the $3\sigma$ significance level. In Table 1, three kinds of time lags are given, $\tau_p$, $\tau_c$ and $\langle \tau \rangle$. Here, $\langle \tau \rangle$ is $(\tau_p + \tau_c)/2$.

It is observed that $\tau_p$ and $\tau_c$ for the optical versus radio are $-59.8^{+59.4}_{-19.8}$ and $-38.1^{+27.6}_{-9.3}$ days, respectively. This result indicates that the optical variation leads to radio variation. $\tau_p$ and $\langle \tau \rangle$ for the optical versus X-ray is $99.3^{+29.8}_{-12.5}$ and $89.5^{+47.5}_{-25}$ days, respectively. This result shows that the X-ray emission zone is upstream of the radio emission region in the jet. Although the result of correlation between X-rays and radio is not significant, the $\gamma$-ray should be upstream of radio emission region in the jet, since the X-ray and $\gamma$-ray has no relative time lags within error.

### 3.3. Location of Emitting Regions

The time lags observed among light curves at various bands can be explained by a perturbation propagating along the jet. The distance between emitting regions at different bands is given by Kudryavtseva et al. [16]

$$D = \frac{\beta_{app} c \Delta T}{(1+z) \sin \zeta} \tag{1}$$

where $\beta_{app}$ is the apparent velocity in the observer frame, $\Delta T$ is the time lag, z is the redshift and $\zeta$ is the viewing angle. For this target, $\beta_{app}$ is given as 0.88 and $\zeta = 4°$. The redshift is set to be z = 0.033 [6]. According to $\tau_p$, $\tau_c$, the corresponding relative distances between

emitting regions at different bands, i.e., $D_p$ and $D_c$, are calculated via Equation (1). Their average $\langle D \rangle = (D_p + D_c)/2$ are given in Table 1.

To ascertain the distance between emitting regions and the jet base, we initially identify the radio core region from the jet base, denoted as $r_{core}$ [27–29]. The $r_{core}$ at the frequency $\nu$ is given by Lobanov [28] and Hirotani [29]

$$r_{core} = \frac{\Omega_{r\nu}}{\nu^{\frac{1}{k_r}} \sin \zeta},$$ (2)

where $\Omega_{r\nu}$ is the core position offset. For Mrk 501, Pushkarev et al. [6] presented $\Delta r_{core,15-8GHZ}$ = 0.279 mas, $\Omega_{r\nu}$ = 3.25 (pc GHz). Assuming the magnetic field energy density is equal to that of particles, we can set $k_r = 1$[29,30]. Then, we calculate $r_{core,15GHz}$ = 3.1 pc via the Equation (2). The correlation between X-ray and radio is marginally at $2\sigma$ significant level in Figure 1, while the correlation between X-ray and optical is close to the $3\sigma$ significant line. Thus, the relative distance between the X-ray and optical is referred to get the X-ray distance from the jet base. We obtain that the distance between the optical emitting region and the jet base is $2.6^{+0.5}_{-0.1}$ pc and the distance between the X-ray emitting region and the jet base is $1.6^{+0.8}_{-0.6}$ pc. As presented previously, the correlation between X-ray and $\gamma$-ray is beyond the $3\sigma$ significance level (see panel (d) in Figure 1), and their relative lags are nearly zero within error. Thus, we consider that the $\gamma$-ray emitting region is the same as the X-ray emitting region. MAGIC collaboration et al. [9] also put forward that a clear correlation without time lag between the X-ray and $\gamma$-ray exist, and express that the X-ray and $\gamma$-ray correlation occurs on both short (weeks) and long (months and years) time scales, which unambiguously indicates a common origin between the emission of these two bands. It can further illuminate the fact that X-rays and $\gamma$-ray are produced by the same particle population.

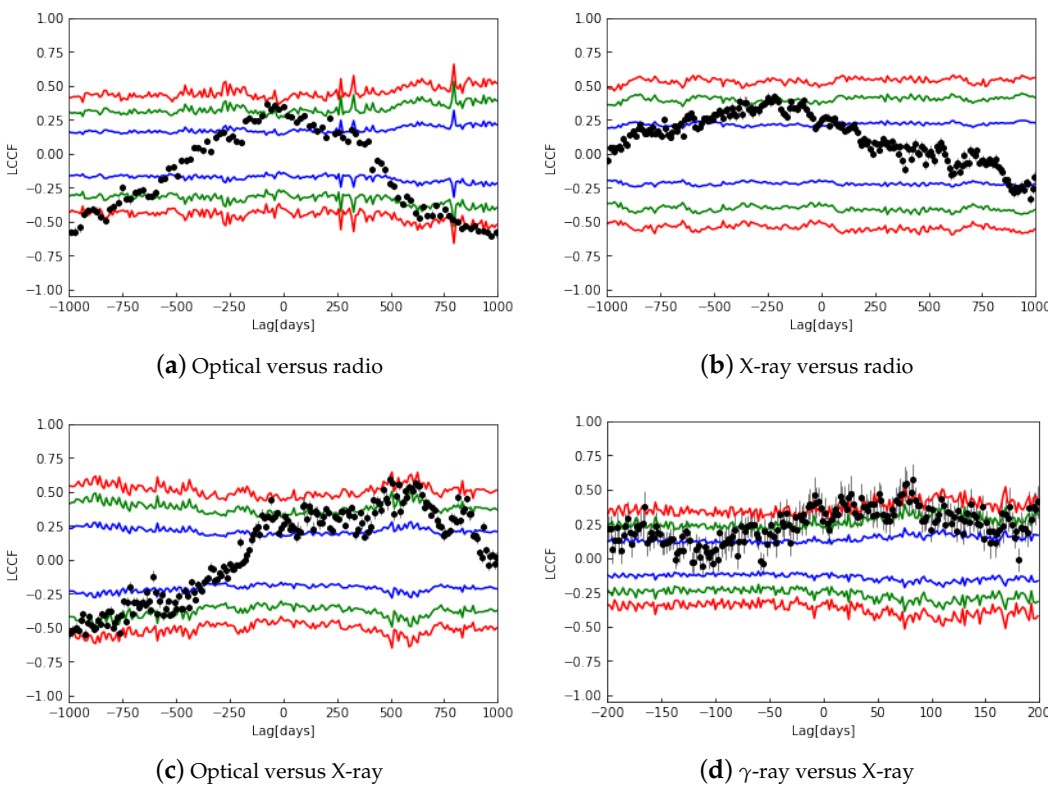

**Figure 1.** The LCCF results for Optical versus radio, X-ray versus radio, Optical versus X-ray, and $\gamma$-ray versus X-ray are presented in panels (**a**–**d**), respectively. The black dots represent the LCCF results, while the significance levels of $1\sigma, 2\sigma, 3\sigma$ are indicated by the blue, green, and red lines, respectively.

Time lags and Relative distances.

**Table 1.** The time lags for panels (a–d) in Figure 1 and relative distances are collected. $\langle D \rangle$ are the average of $D_p$ and $D_c$, respectively. The negative lag value indicates that the former precedes the latter, and that of the distance indicates that the latter is upstream of the former.

| Correlations | Time Lag | | | Relative Distances | | |
|---|---|---|---|---|---|---|
| | $\tau_p$ **(Days)** | $\tau_c$ **(Days)** | $\langle \tau \rangle$ **(Days)** | $D_p$ **(pc)** | $D_c$ **(pc)** | $\langle D \rangle$ **(pc)** |
| Optical vs. radio | $-59.8^{+59.4}_{-19.8}$ | $-38.1^{+27.6}_{-9.3}$ | $-49^{+43.5}_{-14.5}$ | $0.6^{+0.2}_{-0.6}$ | $0.4^{+0.1}_{-0.3}$ | $0.5^{+0.1}_{-0.5}$ |
| X-ray vs. radio | $-219.4^{+10}_{-129.3}$ | $-252^{+21.4}_{-21}$ | $-235.7^{+15.7}_{-75}$ | $2.3^{+1.3}_{-0.2}$ | $2.6^{+0.2}_{-0.2}$ | $2.5^{+0.8}_{-0.2}$ |
| Optical vs. X-ray | $99.3^{+29.8}_{-12.5}$ | $79.6^{+65.2}_{-37.4}$ | $89.5^{+47.5}_{-25}$ | $-1.1^{+0.2}_{-0.2}$ | $-0.8^{+0.4}_{-0.7}$ | $-1^{+0.3}_{-0.5}$ |
| $\gamma$-ray vs. X-ray | $-4^{+0}_{-3.7}$ | $-5^{+1.9}_{-1.5}$ | $-4.5^{+0.9}_{-2.6}$ | $0.04^{+0.03}_{-0}$ | $0.05^{+0.01}_{-0.02}$ | $0.05^{+0.03}_{-0.01}$ |

## 4. Study of Variations

Variability studies are of great significance in exploring the variation mechanisms. In this section, we will analyze the variability phenomena on two different timescales. The optical light curve show three evident activities from MJD 54,700 to 58,150. We use (a), (b), and (c) to mark these three periods at long timescales (years) in Figure 2. Based on the X-ray light curve's flaring states and low states through direct observation, we also select five relative short periods (months), marked with (d), (e), (f), (g), and (h) in Figure 2. In the following, the correlation of flux between different bands, the color index behavior, and variations in polarization can help us to understand the emission and variation mechanisms at these different periods.

### 4.1. Correlation between X-ray and γ-ray

The SED of blazars displays a two bumps structure. It is generally accepted that the first bump is of synchrotron radiation process, and the second is of the self-synchrotron Compton (SSC) process or the external Compton (EC) process, or the combination of them in the popular lepton model. Through the analysis of SED for Mrk 501, we believe that the radiation of X-ray originates from the synchrotron radiation, while the $\gamma$-ray radiation mainly belongs to the second bump. In order to analyze and verify the specific radiation and variation mechanisms, we study the correlations between fluxes at different bands [24]. In the one-zone SSC scenario, the observed fluxes of synchrotron, SSC, and EC, which mainly depend on three parameters, i.e., the particle number density $N_e$, the magnetic field strength $B$, and the Doppler factor $\delta$, are given by Chatterjee et al. [31]

$$
\begin{aligned}
F_{syn} &\sim N_e B^{1+\alpha_X} \delta^{3+\alpha_X}, \\
F_{EC} &\sim N_e \delta^{4+2\alpha_\gamma} U'_{ext}, \\
F_{SSC} &\sim N_e^2 B^{1+\alpha_X} \delta^{3+\alpha_\gamma},
\end{aligned}
\tag{3}
$$

where $\alpha_X$ is the spectral index at X-ray, $\alpha_\gamma$ is that of $\gamma$-ray, and $U'_{ext}$ is the energy density of external photons in the jet comoving frame. To generate data simultaneously, we pair X-ray and $\gamma$-ray fluxes within uncertainty of one day. Figure 3 illustrates the logarithmic relationship between $\gamma$-ray fluxes and X-ray fluxes across the seven periods. The data points of these seven periods were linearly fitted to find the correlations between $\log F_{fl}$ and $\log F_X$. Since there are few pairs in the period (e), we did not show them in Figure 3. The results of linear fitting can be located in Table 2. The fitting results show that there is a weak correlation between them in all periods. However, slopes of the linear fitting in (a), (b), (c), (d), (g), and (h) periods are less than 0.35. At the same time, the correlation between the two is relatively weak, as evidenced by the numerous scattered data points in Figure 3. It is worth noting that although the slope in period (f) is about 2, the error is also large and this result is not significant. The upper and lower limits of the results are both on the high side, and the data points appear significantly scattered in the plot.

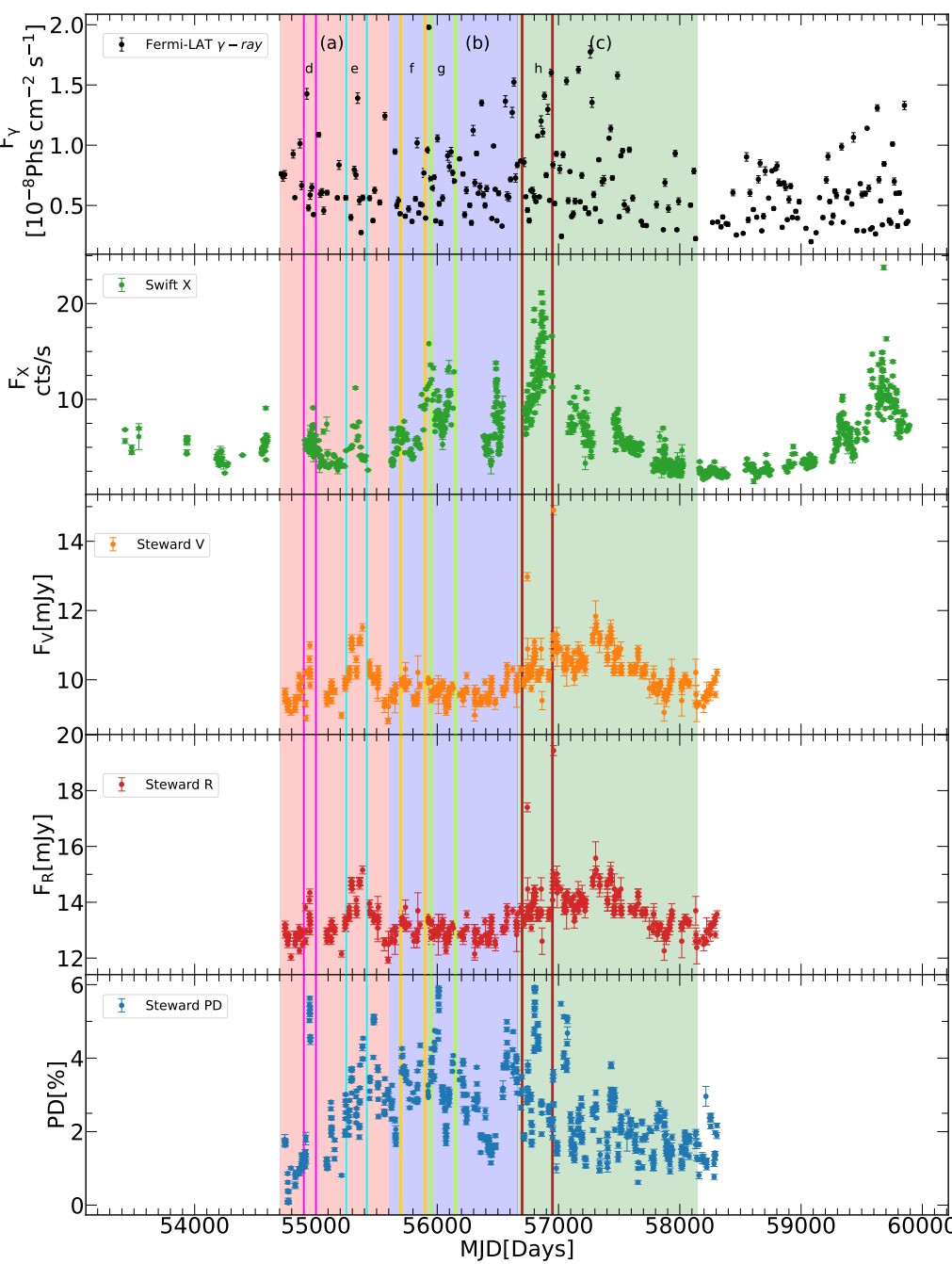

**Figure 2.** From top to bottom panels, the light curves of $\gamma$-ray of 0.9–2.7GeV, X-ray of 0.3–10 keV, optical *V*-band, optical *R*-band and optical PD are plotted, respectively. The red, blue, and green zones are named as period (a), (b), and (c), respectively. Two vertical lines with the same color indicate the periods at short timescales. The magenta, cyan, gold, green-yellow and brown vertical lines corresponds to period (d), (e), (f), (g), and (h), respectively.

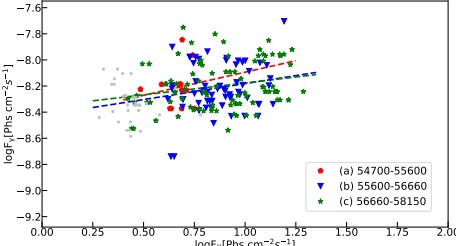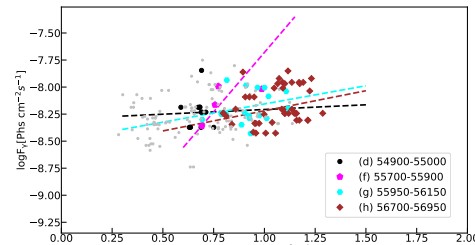

**Figure 3.** The logarithm of the $\gamma$-ray fluxes versus that of the X-ray fluxes is plotted. The color and symbol of the periods (a, b, c, d, e, f, g, h) are indicated in the lower right corner. The periods (a, b, c) are displayed on the left panel, and (d, f, g, h)) is displayed on the right. We set the period to match the color that the red, blue, green, black, spring-green, magenta, cyan, and brown dots are named as period (a), (b), (c), (d), (e), (f), (g), and (h), respectively. Here, the straight line is the result of linear fitting, and the colors used are consistent with the data points. The results of linear fitting can be located in Table 2.

Referencing Table 3 in Shao et al. [24], the slopes can help us study the primary variables. Since the slopes in (a), (b), (c), (d), (g), and (h) are small, the primary variables could not be explained as the particle number density $N_e$ or the magnetic field strength $B$. To verify whether the Doppler factor is the primary variables, the spectral indexes of the X-ray and $\gamma$-ray should be given. For Mrk 501, Mohorian et al. [32] presented the spectral fitting of X-rays. Here, we adopt their best-fit result, i.e., $\alpha_X$ is 1.07. However, the X-ray spectral index is usually varying and using the archival one is a simplification. The range of $\alpha_\gamma$ can be obtained from 0.21 to 1.12 as shown in Figure 4. After calculation, the ranges of $(3 + \alpha_\gamma)/(3 + \alpha_X)$ and $(4 + 2\alpha_\gamma)/(3 + \alpha_X)$ are [0.79, 1.01] and [1.09, 1.53], respectively. Based on the slopes of our linear fits in all periods, the slopes of both do not fall within this range. So, the variation mechanism could not be dominated by the Doppler factor $\delta$ for one-zone model. Based on the above analysis, we can conclude that the primary variables could not be the particle number density $N_e$, the magnetic field strength $B$, or the Doppler factor in both the SSC and EC processes.

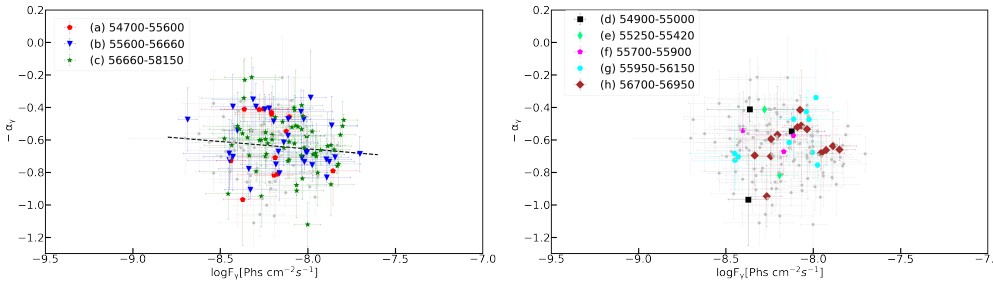

**Figure 4.** The logarithm of the $\gamma$-ray spectral index versus that of the $\gamma$-ray fluxes is plotted. The color and symbol of the periods (a, b, c, d, e, f, g, h) are indicated in the upper left corner. The periods (a, b, c) are displayed on the left panel, and (d, e, f, g, h) is displayed on the right. We set the period to match the color as shown in Figure 3.

*4.2. Color Index and $\gamma$-ray Spectral Index*

The behavior of color gives us the most direct clue to the variation mechanism. In Figure 5, the color $V - R$ versus $V$-band magnitude is plotted. In the whole period and period (a), (b), (c), (d), (e), (f), (g), and (h), the variations of color have the bluer when brighter (BWB) trend. The slope of linear fitting for the whole period in color $V - R$ versus $V$-band magnitude is –0.11 ± 0.006 with Pearson's r = 0.66, and that of (a), (b), (c), (d), (e), (f), (g), and (h) are recorded in Table 2.

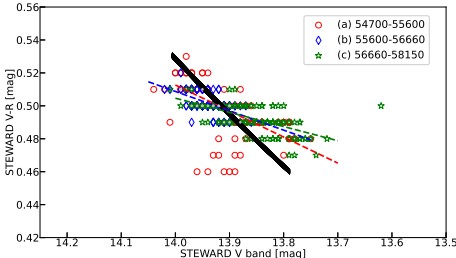 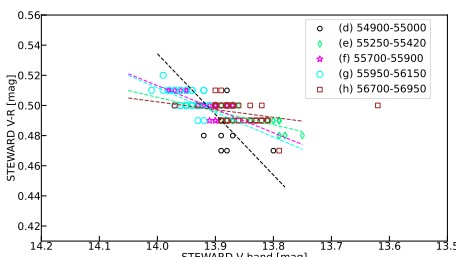

**Figure 5.** $V − R$ versus optical $V$-band magnitude for both the long term periods and short term periods are plotted in the left and right panel, respectively. The color and symbol set is consistent with that used in Figure 3. The black line in the left panel represents our best fitting results of the two-component model, and the other lines are the linear fitting results of data at different periods.

**Table 2.** The outcomes of linear fitting for the $\gamma$-ray flux vs. X-ray flux ((Figure 3)), color index $V − R$ vs. $V$ magnitude (Figure 5), and optical PD vs. optical flux (Figure 6) across eight periods are summarized.

| Periods | $\gamma$-ray Flux vs. X-ray Flux | | $(V − R)$ vs. $V$ | | Optical PD vs. Optical Flux | |
|---|---|---|---|---|---|---|
| | Slope | Pearson's r | Slope | Pearson's r | Slope | Pearson's r |
| a | $0.35 \pm 0.65$ | 0.16 | $-0.16 \pm 0.02$ | $-0.57$ | $7.49 \pm 1.06$ | 0.57 |
| b | $0.24 \pm 0.16$ | 0.19 | $-0.12 \pm 0.02$ | $-0.56$ | $4.19 \pm 1.09$ | 0.31 |
| c | $0.18 \pm 0.08$ | 0.23 | $-0.09 \pm 0.01$ | $-0.67$ | $1.37 \pm 0.49$ | 0.19 |
| d | $0.09 \pm 1.17$ | 0.02 | $-0.4 \pm 0.1$ | $-0.77$ | $8 \pm 2.05$ | 0.76 |
| e | – | – | $-0.09 \pm 0.02$ | $-0.69$ | $2.47 \pm 0.85$ | 0.51 |
| f | $2.20 \pm 2.48$ | 0.46 | $-0.156 \pm 0.03$ | $-0.74$ | $3.21 \pm 0.6$ | 0.8 |
| g | $0.34 \pm 0.33$ | 0.24 | $-0.16 \pm 0.03$ | $-0.60$ | $6.84 \pm 1.88$ | 0.49 |
| h | $0.31 \pm 0.20$ | 0.22 | $-0.05 \pm 0.02$ | $-0.38$ | $1.3 \pm 1.18$ | 0.17 |

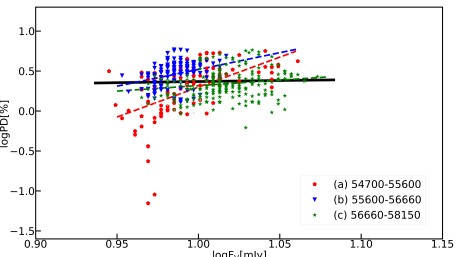 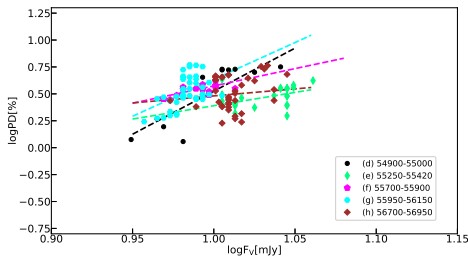

**Figure 6.** logPD versus logF$_V$ is plotted. The color and symbol set is consistent with that used in Figure 3. The color and symbol of the periods (a, b, c, d, e, f, g, h) are indicated in the bottom right corner. The periods (a, b, c) are displayed on the left panel, and (d, e, f, g, h) is displayed on the right. The slopes of the linear fits for various periods in the right panel can be found in Table 2.

The presence of both BWB and redder-when-brighter (RWB) trends in some blazars can be explained by the superposition of both blue and red components, where the red component is of the synchrotron radiation from the relativistic jet, while the blue component comes from the thermal emission from the accretion disk (AD) [33]. Mrk 501 shows strong host galaxy characteristics in the $R$-band [34]. Thus, for Mrk 501, we conclude that the red component is of the host galaxy while the blue component comes from the synchrotron radiation of the jet. The two components model can be used to fit the variation behavior. With this model, the fluxes of the jet and the host galaxy at the $V$ and $R$ bands can be

disentangled. The observed fluxes at $V$ and $R$ bands can be expressed as $F_V = F_V^{jet} + F_V^h$ and $F_R = F_R^{jet} + F_R^h$, where $F_V^{jet}$, $F_V^h$, $F_R^{jet}$ and $F_R^h$ are the jet flux at $V$-band, the host galaxy flux at $V$-band, the jet flux at $R$-band and the host galaxy flux at $R$-band, respectively. From Figure 2, the observed light curves at the $R$ and $V$ bands varies with the same pace. So, we assume that $F_R^{jet} = A \cdot F_V^{jet}$, where $A$ is a constant. We set $F_V^{jet}$ as the variable, $A$, $F_R^h$, and $F_V^h$ as the input parameters. We use the linear model and the Python code (**pymc3**) to predict the best fitted parameters. One disadvantage of the code **pymc3** is that it can only estimate the uncertainty of one input parameter during the Markov Chain Monte Carlo (MCMC) process. So, we first estimate the input parameters manually, and obtain that $A = 0.9$, $F_V^h = 2.1$ mJy, and $F_R^h = 6.25$ mJy. Then, two of the three parameters are set as the fixed input parameters, and the left one is free. After one cycle, we set the obtained best-fitted parameter as fixed input parameter and another parameter as free, and then perform the second cycle of analysis. Finally, three cycles will give the best-fitted and error results for all three input parameters. We also test the permutation of the order of free parameters during the three cycles, and find that the best-fitted results are almost the same within error. In summary, we obtain that $A = 0.89^{+0.01}_{-0.01}$, $F_V^h = 2.13^{+0.04}_{-0.04}$ mJy, and $F_R^h = 6.21^{+0.06}_{-0.06}$ mJy. So, we get that the magnitude of the host galaxy component in the $R$-band is $14.24^{+0.01}_{-0.01}$ mag. We discuss the trend rather than using the MC to obtain an accurate number.

In Figure 4, the correlation between the spectral index $\alpha_\gamma$ and $\log F_\gamma$ is shown. The index of $\gamma$-ray is obtained by the linear fitting of the fluxes at 6 energy bins spaning logarithmically equal between energy 0.3 and 7.29 GeV. In the left panel of Figure 4, the black line represents the result of our linear fit with a slope of $-0.09 \pm 0.01$. This indicates that $\gamma$-ray spectral index exhibits an insignificant soft when bright (SWB) trend. If this weak anticorrelation is true, which means that the $\gamma$-ray becomes soft when bright, it is possible that there is a very hard energy background component, just like the big blue bump causing the redder when bright trend for many FSRQ targets. While in the right panel of Figure 4, the radiation of $\gamma$-ray during (h) period first shows the hard when bright (HWB) trend, then the SWB trend. Its turning point appears in MJD 56919, where $\gamma$-ray flux is $8.4 \times 10^{-9}$ ph cm$^{-2}$ s$^{-1}$, but we consider that the error of this data point is large, and therefore choose the data point of MJD56877 as the turning point, with $\gamma$-ray flux as $8.4 \times 10^{-9}$. This is a very interesting phenomenon that we found, and we will also discuss it in detail in Section 5.

*4.3. Optical Polarization and Optical Flux*

The polarization degree (PD) offers crucial insights into the variation mechanism. The $\log$ PD versus $\log F_V$ is plotted in Figure 6. In the left panel of Figure 6, the linear fittings of optical PD and optical flux in three long timescale period (a), (b), and (c) are plotted. In the right panel of Figure 6, the linear fitting results for five short timescale periods (d), (e), (f), (g), and (h) are plotted. The linear fitting results are summarized in Table 2.

According to our fitting results, it can be concluded that there are positive correlations between the optical PD and optical flux at both the long and short timescales. Wang and Jiang [35] presented an explanation for the positive correlation. The optical emission of the target AO 0235+164 has both the unpolarized disk component and polarized jet component. When the jet component dominates over the disk component, the total PD will increase as the total flux.

For Mrk 501, the correlation between the optical PD and flux can also be explained by the two-component model. In Section 4.2, we mention that Mrk 501 shows strong host galaxy characteristics in the $R$-band. Since Mrk 501 belongs to BL Lacs, we do not consider the contribution of the AD.

The Stokes parameters $I$, $Q$, and $U$ are used to define PD as $PD = \frac{\sqrt{(Q_{jet}+Q_h)^2+(U_{jet}+U_h)^2}}{(I_{jet}+I_h)}$. Since the host galaxy component is unpolarised, the $Q_h$ and $U_h$ can be set as zero. So, the

synthesized PD is written as $\text{PD}_{total} = \sqrt{(Q_{jet})^2 + (U_{jet})^2}/(I_{jet} + I_h)$. In terms of fluxes, one has

$$\text{PD}_{total} = \frac{\sqrt{(Q_{jet})^2 + (U_{jet})^2}}{I_{jet}} \frac{F_V^{jet}}{F_V}, \qquad (4)$$

In this model, when the jet's flux increases, we observe an increase in the total PD. This can explain the positive correlation between optical flux and optical PD shown in Figure 6. In the left panel of Figure 6, the black lines represent our model fitting. Similarly, we use the Python code (**pymc3**) to predict the best fitted parameters. In Section 4.2, we obtain that the host galaxy flux at *V*-band is $F_V^h = 2.13^{+0.04}_{-0.04}$ mJy. When we fit the correlation between optical polarization and optical flux, $F_V^h$ is also set to be 2.13 as a fixed parameter. We set $F_V^{jet}$ as the variable, $\text{PD}_{jet} = \sqrt{(Q_{jet})^2 + (U_{jet})^2}/I_{jet}$ is set as an input parameter with initial value 2.6. After the Python code (**pymc3**) process, we obtain $\text{PD}_{jet} = 2.98^{+0.03}_{-0.03}$. We can find that observed PD are scattered around our best fitted line. Similarly, we discuss the trend rather than using the MC to obtain an accurate number.

## 5. SED Analysis

For Mrk 501, Shukla et al. [36] detected that the photon index increases with the increase of flux for the low-energy $\gamma$-ray radiation in the 0.2–2 GeV energy range during the period from 30 December 2010, to 24 March 2012. This property indicates that the low-energy $\gamma$-ray radiation may be from a different emission region. Thus, the radiation mechanism of $\gamma$-ray for Mrk 501 is relatively complex. We selected three periods for our study: the low flux state (MJD 56,730–56,750), the high flux state (MJD 56,850–56,870), and the post-flare phase (MJD 56,920–56,950). In panel (a) of Figure 7, during the period for MJD 56,730 to MJD 56,750, we observed an excess in the low-energy regime of the $\gamma$-ray. From the light curve, we can infer that, during this period, the $\gamma$-ray emission was in a relatively low state of activity. This observation leads us to consider the existence of a soft component that dominates the low-energy emission of the $\gamma$-ray at low state. To verify this aspect, we also select three other periods with low activity in the $\gamma$-ray band. The results are shown in Figure 8. From this figure, we can see that there is indeed a plateau present in the low-energy regime of the $\gamma$-ray emission.

When studying the origin of the hard X-ray excess of the HSP BL Lac object Mrk 421, Chen [37] proposed that the physical mechanism responsible for the excess can be attributed to the Spine/Layer jet, as inferred from the fitting of the broadband SED. The soft plateau observed in the $\gamma$-ray emission during the low-activity state of Mrk 501 can also be attributed to the Spine/Layer jet. According to MacDonald et al. [38], the sudden increase in the energy density of external seed photons (produced from the layer) can cause an orphan flare in the $\gamma$-ray band when a faster-moving spine (emission zone) passes through a slower-moving (or steady) layer/ring, since the spine electrons upscatter these layer photons via IC scattering. Mrk 501 exhibits an interesting feature known as a misaligned AGN, with a significant orientation difference between its inner parsec-scale jet and the kpc-scale jet structure [38]. Recent observations of the kinematics of the parsec-scale jet in Mrk 501 have revealed a fascinating phenomenon, i.e., the outer portion of the jet exhibits a significant drift perpendicular to the inner jet, resulting in a gradual straightening of the jet over time. Additionally, based on the radio observation, Britzen et al. [39] suggests the presence of a spine-sheath structure within the jet. To provide a specific explanation for the existence of the spine-layer structure, we employ a Gaussian component to represent the layer jet and investigate its flux variations. In other words, we study how the layer jet contributes to the occurrence of excess low-energy gamma-ray emissions.

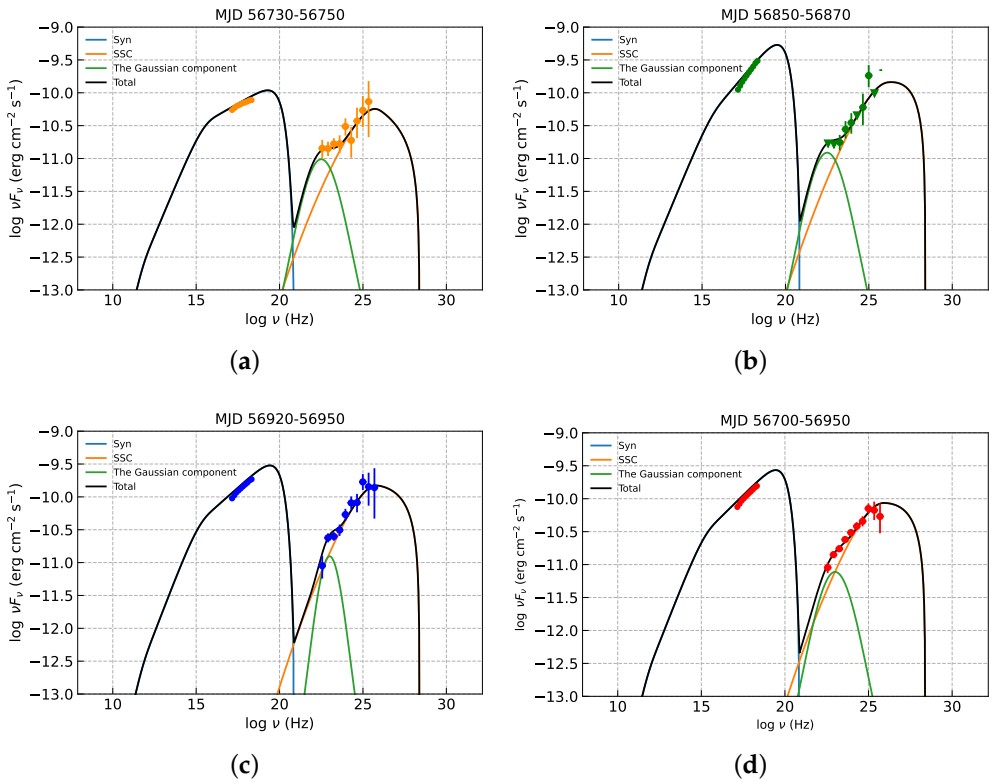

**Figure 7.** The broadband SED of Mrk 501 are given. The panel (**a**–**d**) represents that the SED of Mrk 501 derived with data from MJD 56,730 to MJD 56,750, MJD 56,850 to MJD 56,870, MJD 56,920 to MJD 56,950, MJD 56,700 to MJD 56,950, respectively. The corresponding model parameters are shown in Table 3.

**Table 3.** Parameters of the one-zone SSC plus Gaussian model.

| Periods | MJD 56,730–56,750 | MJD 56,850–56,870 | MJD 56,920–56,950 | MJD 56,700–56,950 |
|---|---|---|---|---|
| Flux State | Low Flux State | High Flux State | Low State | Mean |
| $N_0(cm^{-3})$ | 20 | 20 | 10 | 10 |
| $\gamma_{min}$ | $10^3$ | $10^3$ | $10^3$ | $10^3$ |
| $\gamma_b$ | $10^5$ | $1.2 \times 10^5$ | $7.5 \times 10^4$ | $7.5 \times 10^4$ |
| $\gamma_{max}$ | $2 \times 10^7$ | $2 \times 10^7$ | $2 \times 10^7$ | $2 \times 10^7$ |
| $p_1$ | 1.78 | 1.77 | 1.67 | 1.7 |
| $p_2$ | 2.71 | 2.4 | 2.55 | 2.5 |
| B(G) | 0.01 | 0.01 | 0.01 | 0.01 |
| R(cm) | $10^{17}$ | $10^{17}$ | $10^{17}$ | $10^{17}$ |
| $\delta$ | 12 | 12 | 12 | 12 |
| $\mu$ | 22.5 | 22.5 | 23 | 23 |

Mrk 501 belongs to a BL Lac object. The SED fitting of BL Lacs usually considers the one-zone SSC model [11,40]. So, we first consider the one-zone SSC model of the Spine component to fit the normal high energy bump at the $\gamma$-ray band. Considering previous works, we also introduce a Gaussian component in addition to the one-zone SSC model with the aim of explaining the plateau in the low state of $\gamma$-ray. In the one-zone SSC model, a broken power-law of electron energy distribution is assumed, i.e.,

$$N(\gamma) = \begin{cases} N_0\gamma^{p_1} & \gamma_{\min} \leq \gamma \leq \gamma_b, \\ N_0\gamma_b^{p_2-p_1}\gamma^{-p_2} & \gamma_b < \gamma \leq \gamma_{\max} \end{cases} \tag{5}$$

where $\gamma_{\min}$ and $\gamma_{\max}$ are the minimum and maximum electron Lorentz factors, and $N_0$ is the normalization of the particle number density. The emission region is assumed to be a homogeneous column. The size of the column can be estimated from the variation time via $R < c\delta\Delta t/(1+z)$, where $c$ is the speed of light and $\Delta t$ is the minimum variation time [41]. Here, we estimate $\Delta t$ via the flux doubling time, during which the flux undergoes a change of factor 2 or more in consecutive time intervals. Its formula is expressed as

$$t_d = \left| \frac{(f_B + f_A)(t_B - t_A)}{2(f_B - f_A)} \right|, \tag{6}$$

where $t_A$ and $t_B$ are the two successive times with fluxes $f_A$ and $f_B$, respectively. The minimum variation time $\Delta t$ is determined by estimating the smallest value of $t_d$ from all consecutive time intervals during one $\gamma$-ray flare. In Figure 2, we consider the $\gamma$-ray flare during the period (h), and find that $\Delta t$ is about 8.4 days. In detail, the related parameters are $t_A = 56{,}856$, $t_B = 56{,}863$, $f_A = 1.2 \times 10^{-8}$ph cm$^{-2}$s$^{-1}$ and $f_B = 4.94 \times 10^{-9}$ph cm$^{-2}$s$^{-1}$. According to $R < c\delta\Delta t/(1 + z)$, we can obtain $R < 2.5\times 10^{17}$ cm. Therefore, we fix the emission zone size to be $R = 10^{17}$ cm and the Doppler factor to be $\delta = 12$. This is very close to the parameters given by Abdo et al. [40] to fit the broad-band SED of Mrk501. Furthermore, we assume that the minimum energy of electrons is $\gamma_{min} = 10^3$, the maximum energy of electrons is $\gamma_{max} = 2 \times 10^7$ and magnetic intensity is 0.01 (G), which helps to further refine the fitting procedure.

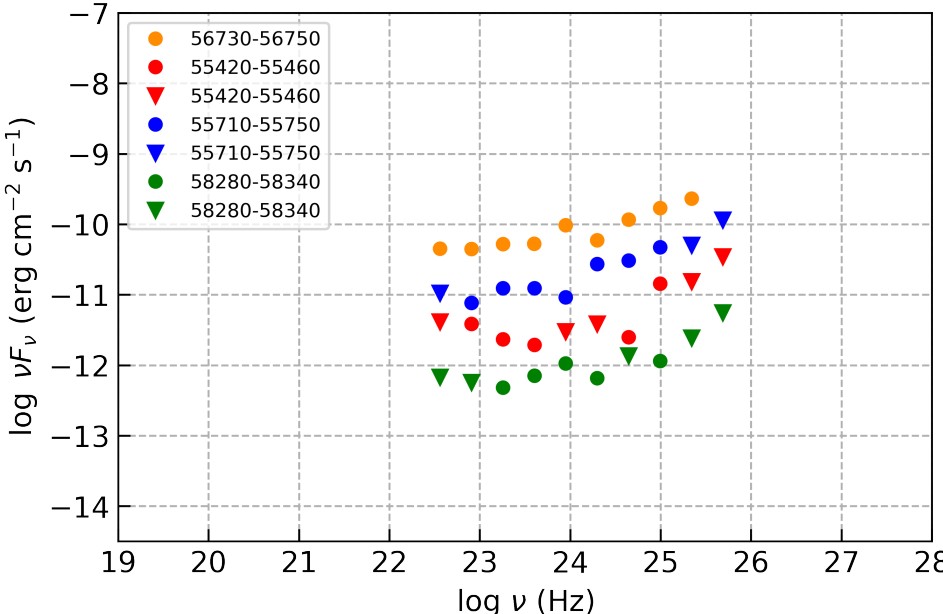

**Figure 8.** The SED of $\gamma$-ray band in low activity states is expressed. The dots represent energy data points and the inverted triangles represent their upper energy limits. In order to clearly distinguish the data points of different periods, we choose to add a constant to their vertical coordinates. We add 0.5, $-0.5$, $-1.1$ to the vertical coordinates of the data points of dark-orange, red, and green colors, respectively. Blue data points are not changed.

For the Gaussian component, we use the function $f_G = ae^{-(x-\mu)^2/2\sigma^2}$, where $a = 1/\sigma\sqrt{2\pi}$. In Figure 7, the Gaussian function components are plotted. We implement the fitting by adjusting the parameter $\mu$, $\sigma$ and stretching the Gaussian function. For both panel (a) and (b) of Figure 7, $\mu$ is equal to 22.5. The amplitude of the Gaussian component of panel (b) is higher than that of (a). However, in panel (c) of Figure 7, we obtain that $\mu$ is equal to 23. We adjust feasible values of free parameters, and find the best fitted results manually. The results of our fit are plotted in Figure 7, and the fitted parameters are given in Table 3.

In our SED fit, as seen by the fitted parameters in Table 3, the broken energy $\gamma_b$ of electrons first increases from the low to the high flux state, and then decreases after MJD 56850–56870. In addition, we find that the peak frequency of the Gaussian component is increasing throughout the three periods in Figure 7. MAGIC Collaboration et al. [11] also conducted SED study using a one-zone SSC model from Tavecchio et al. [42] on the X-ray active period from MJD 56854 to MJD 56869. They attribute most of the broadband flux variations during this period to $\gamma_b$, which could change due to the electron acceleration and cooling processes in the shock in jet model. This viewpoint has also been supported in our model.

The trend transition in period (h), as shown in the right panel of Figure 4, could be explained well in this one-zone SSC plus a Gaussian component model. From the low flux state (MJD 56,730–56,750) to the high flux state (MJD 56,850–56,870), $\gamma_b$ increases, while the Gaussian component is almost invariant. This means that the SSC component increases prominently compared to the Gaussian component, leading to the HWB trend. Then, at the period MJD 56,920–56,950, $\gamma_b$ decreases while $\mu$ of the Gaussian component increases. This leads to the SWB trend naturally. These variation phenomenon could be explained with the spine-layer jet model. Since the spine of jet will contribute the major SSC radiation, the layer component may contribute the soft background (the Gaussian component). When the shock in jet propagates from the spine to the layer, the radiation of the layer component lags that of the spine component. The trend transition occurs naturally in the manner. The spine-layer jet is one possible scenario to explain the soft plateau during the low state of $\gamma$-ray. There are other scenarios that could also explain the soft excess. Abdo et al. [40] observed that the low-energy segment of the formed electron energy distribution, dominating the production of observed $\gamma$-rays below a few GeV in the study of Mrk 501, appears to exhibit low and relatively slow variability. Simultaneously, when fitting the Spectral Energy Distribution (SED) by considering contributions from different segments of the electron energy distribution to fit the GeV–TeV $\gamma$-ray spectrum of Mrk 501, we speculate that the observed plateau may also arise from the dominance of the low-energy segment of the electron energy distribution. Further work is needed to confirm the discovery of this phenomenon.

## 6. Conclusions

In this study, we gathered multi-wavelength light curves of Mrk 501, encompassing $\gamma$-ray, X-ray, optical, optical PD, and radio data. A detailed LCCF analysis was conducted, leading to the determination of the locations of the optical, X-ray, and $\gamma$-ray emitting regions. We investigated the variation behaviors over three long-term and five relatively short-term periods. Our analysis included the examination of correlations between X-ray flux and $\gamma$-ray flux, optical color index, and optical magnitude, as well as optical flux and optical PD. The primary conclusions are summarized as follows:

- Based on the LCCF results, we observed that the distance between the jet base and the optical emitting region is $2.6^{+0.5}-0.1$ pc, while for the X-ray emitting region, it is $1.6^{+0.2}-0$ pc. We also get that the $\gamma$-ray emitting region is most likely the same as the X-ray emitting region and X-rays and $\gamma$-ray emitting are produced by the same particle population. The X-ray, $\gamma$-ray, and optical emitting regions along the jet are situated upstream of the radio core region. Similar research has also been conducted by Bhatta [43], Acharyya and Sadun [44];

- We analyzed the correlation between X-ray and $\gamma$-ray. We get the slope of $\log F_X$ vs. $\log F_\gamma$ in different periods. The results show that the SSC process modulated by Doppler factor can be ruled out as the variation mechanism for the $\gamma$-ray radiation in the one-zone SSC scenario. For the variation of color index and the optical PD, we found the BWB trend and the positive correlation between the optical PD and flux, which could be explained by the two-component model. Based on this model, We obtain that the magnitude of the host galaxy component in the $R$ band is 14.23

magnitude. This is consistent with the values measured by Nilsson et al. [45] with an aperture diameter of 5 arcsec;

- We detected the presence of a soft component in the low-activity state of $\gamma$-ray radiation. The broadband SED for the period from MJD 56,700 to MJD 56,950 is fitted by using the one-zone SSC plus the Gaussian component model. The Gaussian component may originate from the layer of the jet.

**Author Contributions:** writing—original draft preparation, L.L.; writing—review and editing, Y.J.; software, J.D.; investigation, Z.C. and C.M. All authors have read and agreed to the published version of the manuscript.

**Funding:** This work has been funded by the National Natural Science Foundation of China under grant no. U2031102, and by the Shandong Provincial Natural Science Foundation under grant no. ZR2020MA062.

**Data Availability Statement:** Data from the Steward Observatory spectropolarimetric monitoring project were used. This program is supported by Fermi Guest Investigator grants NNX08AW56G, NNX09AU10G, NNX12AO93G, and NNX15AU81G. The data used in this research are available at Fermi Science Support Center, NASA's HEASARC webpages with the links given in the article.

**Conflicts of Interest:** The authors declare no conflict of interest.

## Notes

1. https://fermi.gsfc.nasa.gov/cgi-bin/ssc/LAT/LATDataQuery.cgi (accessed on 1 January 2024).
2. https://pypi.org/project/easyFermi/ (accessed on 1 January 2024).
3. http://www.swift.psu.edu/monitoring/ (accessed on 1 January 2024).
4. http://james.as.arizona.edu/~psmith/Fermi/ (accessed on 1 January 2024).
5. http://www.astro.caltech.edu/ovroblazars/ (accessed on 1 January 2024).

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
