# Peer review of "Unveiling the Emission and Variation Mechanism of Mrk 501: Using the Multi-Wavelength Data at Different Time Scale"

_universe, doi:10.3390/universe10030114_

Round 1
Reviewer 1 Report
Comments and Suggestions for Authors
See the report.

Comments on the Quality of English LanguageRequires moderate edditing
Author Response
Thank you for your suggestion, our response is in the docx file

Reviewer 2 Report
Comments and Suggestions for Authors
Useful approach and interesting results. Worth to be published.
Author Response
Thank you for your suggestion.
Reviewer 3 Report
Comments and Suggestions for Authors
This is a well structured manuscript that tries to analyze a nice long-term MWL data set. While there are already other studies covering similar time ranges and conclusions, it for the first time combines long-term optical with X-ray data and provides a valuable cross check for previous studies. However, some of the methods have to be better explained, improved or cross checked, as I explain in more detail below. But I believe that with a careful consideration of the comments given, the manuscript can reach a valuable contribution to the understanding of Mrk 501.
Here a list of the major comments on methods or results that should be checked more carefully (minor comments on rephrasing or missing citations are given at the end). For convenience, I attach the report also as a pdf.
Section 2:
Fermi-LAT Analysis:
The analysis is a bit outdated and should be updated to match the data set used. Additional, some analysis settings are not stated:
-
The 4FGL catalog only goes to 2016, the data to 2022. Please use the most up-to-date 4FGL-DR3 catalog.
-
The ScienceTools package used is from 2018, please use a more modern version.
-
Please match your background models correctly. gll iem v07.fits should go with 𝑃8𝑅3_𝑆𝑂𝑈 𝑅𝐶𝐸_𝑉3, see https://fermi.gsfc.nasa.gov/ssc/data/access/lat/BackgroundModels.html
-
L89/90: How exactly is this done? Did you optimize on the LAT data by checking different energy ranges and seeing which gives you the best results? Then you have to take into account trial factors with a trial for each energy range you use. And then propagating the decrease in significance you get from the trial factor to the flux uncertainty. Therefore, most likely you will remove all advantages you have by choosing this range. Instead of this fine tuning, it might be safer to just use a standard energy range. Please either use a standard energy range or take into account the trial factor of your fine-tuning correctly.
-
L91: 473 data points * 7days = 3311 days -> 9 years. But in L82 you write 14 years. Where does the mismatch come from?
-
L92: Please explain better why only this MJD period is analyzed separately with easyFermi. Is it because you use easyFermi to construct SEDs?
-
Please state the following settings of your LAT analysis to allow reproduction:
-
Which evt class and evt type do you use?
-
Which sources of the 4FGL catalog are included in your model for the fit? All? Or do you remove some after a prefit, which are not significant during the considered time period?
-
Which source parameters are fixed and which are left free during the fit?
-
Which spectral model do you consider for Mrk 501?
Swift-XRT analysis:
-
Why do you only go until 2020? If Fermi is to 2022, please extend the XRT analysis also until 2022. Especially since otherwise you consider the same time range as Abe et al 2023 for the correlations.
-
L99/100: “a radius of 40 and 50 pixels” and L104 “with a minimum of 20 and 10 photons” - so you do two separate analyses? Which one is the one then used in the paper and why do you have two versions?
-
You only use WT mode data, not from PC mode. Please state why and how much data you are excluding because of it.
-
L106: which energy bands do you choose and why 10? Later on you only use the full energy range for the LC and correlations. Do you mean you bin the SEDs in 10 energy bins? Please explain better in the text.
Photometry and polarization data:
-
Please mention that you do not do any host galaxy correction. In case you do the correction, please mention how.
Radio data:
-
L81: “from public data archives” either here or in the radio section please also cite from which data archive you took the data from. I guess it was Abe et al 2023 because you use the exact same dates.
-
L354: If possible, please add to the Data Availability Statement “This research has made use of data from the OVRO 40-m monitoring program [13] supported by private funding from the California Institute of Technology and the Max Planck Institute for Radio Astronomy, and by NASA grants NNX08AW31G, NNX11A043G, and NNX14AQ89G and NSF grants AST-0808050 and AST- 1109911.”
Section 3:
LCCF analysis:
-
L127: Why do you only state the spectral PSD slopes of X-ray and radio? What about optical and Fermi? Same for the number of data points.
-
- L128: The number of data points you get are the same as in the real LCs? Do you apply the same time sampling as in the real LCs to the simulated ones? If not, please do so and check the results again.
-
- L134: How do you define the time lags? If a time lag is positive, is the LC at higher energies lagging the one at lower energies? Or the opposite? Or do you mix it. This is important to interpret the results.
-
- L140: why do you only choose to discuss the results of these combinations of LCs. Did the other not show interesting peaks? At least radio to gamma-ray should show something since you discuss it later in L 153. Please add the results you get for it.
-
L143: why do you use a binning of 10 days ? Why not 7 days as is your Fermi binning? Or is the optical/radio/Xray binning/sampling bigger than 7 days on average? Please specify why in the text.
- And why do you use an even shorter one (2 days) for Fermi to X-Rays if Fermi is binned in 7 days. This should be redone within 7 days. -
L153: “significant”. How do you define significance? Usually it is >3sigma and then also your results before are not significant. And as mentioned before, please also show the gamma to radio LCCF somewhere, if you discuss it here.
Figure 1:
-
The Fermi-LAT LC looks more fluctuating than usually for Mrk 501. Please provide an Xcheck of your methods using the standard Fermi-LAT energy range of 0.1-100GeV with the public LC of the Fermi light curve repository: https://fermi.gsfc.nasa.gov/ssc/data/access/lat/LightCurveRepository/source.html?source_name=4FGL_J1653.8+3945
-
The Swift-XRT LC is usually given in erg cm-2 s-1. Please convert it.
-
Caption: Radio 15GHz is mentioned in the caption but now shown and Xray is mentioned between R-band and PD even though it is the second panel from the top.
Section 4:
-
Please specify better why and how the time intervals a-h are selected (L180ff). For now it is a bit confusing what is the motivation behind.
-
Also please quantify how you select the time intervals, by eye from the LCs is not a valid method. You can e.g. fit Gaussians with a constant background to select flares/low states or you can use flux levels (e.g. low state for flux < XX, intermediate with XX < flux < XX, high state with flux > XX).
Section 4.1:
-
Keep in mind that EC is usually not considered for BL Lacs since there are not strong broad emission lines hinting towards possible seed photons.
-
L188: How do you pair the 7day binned gamma-ray LC with the X-ray in 1 day bins? Do you take the gamma-ray one multiple times ? Please check your results also with a 7day binning to match the two LCs.
-
L195: Please specify how you define significance here.
-
L199 : -> “as the particle number density 𝑁𝑒 or the magnetic field strength 𝐵 assuming a one-zone SSC model to explain the full broadband SED. “ Because if you have multiple emission regions or a spine sheath as you consider later, this analysis does not hold anymore. So please specify it.
-
L201: The spectral index in the X-rays can be very variable for Mrk 501. So please do not use data from a different data set, especially since XMM (as in Mohorian) samples far less than Swift-XRT. Please get the spectral index from your own Swift analysis and obtain a range of spectral indices (min, max and mean) as you consider for Fermi-LAT.
-
L202: The Fermi-LAT indices seem off, 0.21 is far too small. How did you obtain them? Did you consider a Power law or a log parabola model? If the second, please make sure to freeze the curvature parameter to the catalog value to study the variation of alpha correctly.
-
L205-207: Again, please write concretely that this only refers to a one-zone SSC scenario and also (either here or in the conclusions, L341) what are the conclusions from this? Are there other primary variables possible? Or does it mean that we have a more complicated model than a one-zone one? same in abstract, add that the conclusion for the Doppler factor not favored it for a one-zone SSC.
Section 4.2:
-
L231ff: The errors you get from this method, where you only leave one parameter free at a time, are not completely trustable because you do not take into account correlations between the parameters. So you can use them to get trends, but not to claim any significance etc because then you would need to take the dependencies of the different parameters into account. Please discuss this somewhere.
-
L235: You can also get the host galaxy flux from Nikson et al 2007 using the aperture and FWHM of the Steward data, which is the traditional method. Please compare your results to the traditional method.
-
L240: Please be very careful with the HWB trend in (h). It is only one outlier with a huge error bar, so most probably not significant. Please quantify the trend before making conclusions on it (as you do in L315).
-
L258: Same as before with freeing only one parameter at a time and not using error propagation: Please discuss that the accuracy of the results is affected by it and they should be treated as trends.
-
Also you can correct the PD for the host galaxy flux (see e.g. Weaver et al 2020). Please compare your method with the usual host galaxy correction.
-
How can you get a constant PD_jet if it is dependent on a constant host galaxy contribution and a variable jet flux contribution? Should it not also vary with the variable F_v^{jet}?
Section 5:
-
The time ranges and epochs in Figure 7 have to be introduced better. How are they chosen and why?
-
L269: How do you test if there is a plateau and that it is only during low-activity? Please compare it to gamma-ray spectra during other states as well.
-
L275ff: how are orphan flares important to explain the gamma-ray plateau for low-states? Consider removing this sentence to not confuse the reader.
-
The model: You are mixing a physically meaningful model with a descriptive Gaussian on top, which needs more reasoning why this is okay:
-
Please provide proof/citations that the spine sheath model would lead to an additional Gaussian in the Fermi range (as you claim in L 318)
-
Please check if an one-zone SSC model can also explain the SEDs and compare it to your model to see if the added Gaussian is preferred or a simple one-zone model also works.
-
L323-325: The host galaxy is too far away from the jet to provide enough seed photons for EC. Please remove the sentence (same for L348/349)
-
Figure 7: The Gaussian is not constrained because there are only ULs in the relevant part of the Fermi SED. Please try a different binning for the Fermi SED.
-
Table 3: From here it seems like also B and gamma_max are fixed. Please mention that in the text as well.
Conclusions:
In general you have to put your results more in the context of what has been found before.
-
For the LCCF: Many correlations you found are not new and especially not your conclusions on which emission regions coincide and which don’t. Please put your results in context in L336 comparing it to the results from Bhatta 2021, Abe et al 2023 and Acharyya and Sadun 2023.
-
L344: Put in context by comparing it with the traditional value from Nilson et al 2007
Minor comments:
-
L15: It might be good to also mention the definition of BL Lacs as a subclass of blazars.
-
L18: remove (LOS). Line of sight is only used here, so no need to define is as an acronym
-
L18 to 22: Citations missing to where you get these information from.
-
L28: Citations missing to LSP, ISP, HSP definitions. E.g. Abdo et al 2010
-
L34-L36, this depends on how you define “innermost part of the jets”. Please specify better and add citations.
-
L57-64: the 3TeV bump is not used in the paper or modelling afterwards. Therefore, I would remove this part of the paragraph or at least shorten it.
-
L69: Define what you mean with primary variable.
-
L80: Swift -> Swift-XRT.
-
L138: citations missing fo FR and RSS methods
-
Formula 1: Remove citation for Max-Moerbeck et al [15]. You use the formula from 14 anyway and the Max-Moerbeck paper having a similar one is your [17], but it is dependent on the Doppler factor, which yours is not.
-
L166: Please provide both calculations. Once using your X-ray to optical and once the X-ray to radio. Because both correlations are <3 sigma and therefore it is dangerous to values one higher than the other.
-
Section 4.1: Would not call it correlation since already with the LCCF you investigate it. Maybe slope of correlations? Same in L191 and L330, L339.
-
L186: SSC and EC need citations.
-
L202: Figure 5 is mentioned before Figure 4 in the text, please swap the two Figures.
-
L226: citation or link to pymc3 missing.
-
L255: unit of 2.13 is missing
-
Figure 6: Define black line in caption
-
L261 and 265: remove low-energy, Fermi-LAT are high-energy gamma-rays.
-
L263: “different emission region”. Different to what? Please specify
Comments on the Quality of English Language
The quality of English of the manuscript is fine. There are a few sentences that do not make sense, but they can be considered during the proofs.
Author Response

(The authors gave the same response as above.)

Round 2
Reviewer 3 Report
Comments and Suggestions for Authors
Thank you for the revised version. I understand that it had to be done on a tight
timeline and value the improvements made in this time and how my comments were addressed. Many comments have been addressed to my satisfaction.
However, not all comments were considered and some others still need a bit more rework. For the open ones, I've provided replies in the attached pdf.

Comments on the Quality of English LanguageThe English is okay and only single sentences might need more revision during a language edit.
Author Response
Thank you for your suggestion, our response is in the file.
